# User Distribution Mapping Modelling with Collaborative Filtering for Cross Domain Recommendation

## ABSTRACT

User cold-start recommendation aims to provide accurate items for the newly joint users and is a hot and challenging problem. Nowadays as people participant in different domains, how to recommend items in the new domain for users in an old domain has become more urgent. In this paper, we focus on the *Dual Cold-Start Cross Domain Recommendation* (*Dual-CSCDR*) problem. That is, providing the most relevant items for new users on the source and target domains. The prime task in Dual-CSCDR is to properly model user-item rating interactions and map user expressive embeddings across domains. However, previous approaches cannot solve Dual-CSCDR well, since they separate the collaborative filtering and distribution mapping process, leading to the error superimposition issue. Moreover, most of these methods fail to fully exploit the cross-domain relationship among large number of non-overlapped users, which strongly limits their performance. To fill this gap, we propose User Distribution Mapping model with Collaborative Filtering (**UDMCF**), a novel end-to-end cold-start cross-domain recommendation framework for the Dual-CSCDR problem. **UDMCF** includes two main modules, i.e., *rating prediction module* and *distribution alignment module*. The former module adopts one-hot ID vectors and multi-hot historical ratings for collaborative filtering via a contrastive loss. The latter module contains overlapped user embedding alignment and general user subgroup distribution alignment. Specifically, we innovatively propose unbalance distribution optimal transport with typical subgroup discovering algorithm to map the whole user distributions. Our empirical study on several datasets demonstrates that **UDMCF** significantly outperforms the state-of-the-art models under the Dual-CSCDR setting.

## CCS CONCEPTS

• **Information systems** → **Recommender systems**.

## KEYWORDS

Recommendation, Cross Domain Recommendation, Domain Adaptation, Optimal Transport

**ACM Reference Format:**
Anonymous Author(s). 2023. User Distribution Mapping Modelling with Collaborative Filtering for Cross Domain Recommendation. In *WWW '24: The 46th International ACM SIGIR Conference on Research and Development*

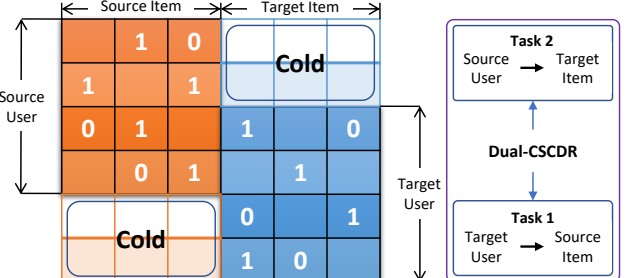

**Figure 1: The problem of Dual Cold-Start Cross Domain Recommendation (Dual-CSCDR).**

*in Information Retrieval, July 23-27, 2023, Taipei, Taiwan.* ACM, New York, NY, USA, 10 pages. https://doi.org/10.1145/1122445.1122456

## 1 INTRODUCTION

Recommender systems become more and more attractive with the big data explosion in recent years. With the advent of digital era, more and more users participant in multiple domains (platforms) for different purposes, e.g., watching movies on *Netflix* and buying books on *Amazon* [9, 52]. Meanwhile, how to provide the most relevant new items for users across domains has become a hot topic. Therefore, the Cross-Domain Recommendation (CDR) has emerged to utilise and exploit useful knowledge for achieving promising solution on the cold-start recommendation [23, 26, 56, 57]. CDR models can transfer sharing patterns inherited from multiple domains to enhance the model performance for better results. However, most of current CDR models always assume that all users or items are strictly overlapped which limits their potentials, especially for the cold-start CDR tasks with only few overlapped users/items.

In this paper, we concentrate on the *Dual Cold-Start Cross Domain Recommendation* (Dual-CSCDR) problem, that is, providing the most relevant items in a domain for new users in the other domain in a bi-directional way (e.g., cold-start source users without historical interactions in the target domain) without other auxiliary representations. The Dual-CSCDR problem popularly exists in practice with two main tasks, i.e., (1) recommending source items to target users and (2) recommending target items to source users which have been shown in Fig.1. The prime task of Dual-CSCDR problem is bridging and mapping users' preferences across domains for reducing the domain bias and discrepancy.

Although there have been previous studies on the CSCDR problem, these models cannot solve Dual-CSCDR well. On one hand, current CSCDR models always separate the procedure of collaborative filtering and transferable bridge mapping. As a result, the knowledge across domains will not be fused together during the collaborative filtering step, which leads to the error superimposition problem and increase the final recommendation error [60]. On the other hand, most of the CSCDR models cannot fully explore

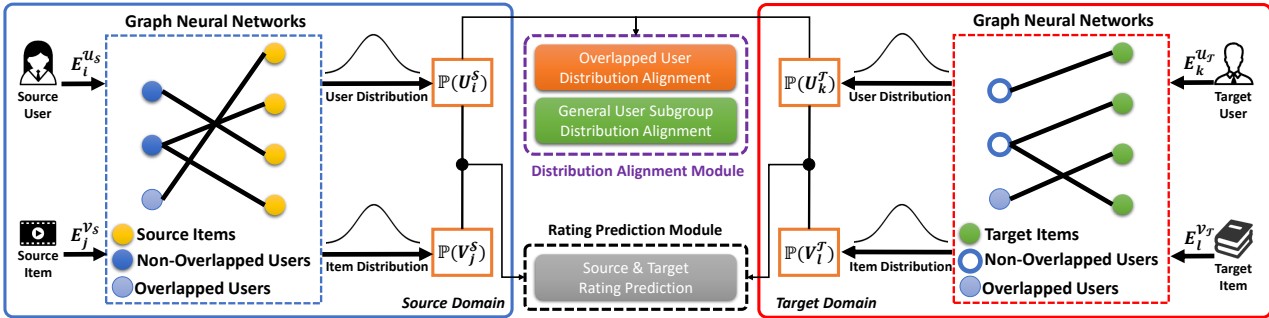

**Figure 2: The main framework for the proposed UDMCF.**

the useful representations behind the non-overlapped users. As a consequence, a large amount of information will be neglected, which leads to model degradation, especially when the number of non-overlapped users are much more than the overlapped users. Furthermore, they fail to fully exploit and align the embedding structure and probability distribution for the whole user feature space. Therefore, current CSCDR models cannot solve these challenges and lead to limited model performance.

To address the aforementioned issues, in this paper, we propose User Distribution Mapping with Collaborative Filtering (**UDMCF**) model, a recommendation framework for the Dual-CSCDR problem. We propose two modules in **UDMCF**, i.e., *rating prediction module* and *distribution alignment module* for better modelling user/item embeddings and transferring relevant information across domains. In the rating prediction module, we utilize the one-hot ID vector to generate user and item preference distribution via collaborative filtering on graph neural network. The distribution alignment module tends to reduce the domain discrepancy between the source and target domains. We first propose distribution alignment for both overlapped and non-overlapped users. Specifically, we innovatively propose latent subgroup distribution alignment with user subgroup distribution measurement and alignment. As a result, our proposed **UDMCF** can be trained end-to-end through modelling user/item preferences and mapping users with similar characteristics.

We summarize our main contributions as follows: (1) We propose a novel end-to-end training framework, i.e., **UDMCF**, for the Dual-CSCDR problem, which contains rating prediction module and distribution alignment module. (2) To our best knowledge, this is the first attempt in literature to align both the overlapped and non-overlapped users embedding distribution across domains based on proposed latent distribution alignment via the unbalanced distribution optimal transport. (3) Extensive empirical studies on four datasets demonstrate that **UDMCF** significantly improves the state-of-the-art models, especially under the Dual-CSCDR setting.

## 2 RELATED WORK

**Traditional Cross Domain Recommendation.** Traditional Cross Domain Recommendation (CDR) models aim to provide a promising solution for tackling the data sparsity in the target domain. These methods leverage the relative rich auxiliary information to enhance the performance in the sparser domain. Existing CDR works on this are mainly of two types, i.e., *rating-based* methods and *supplement-based* methods [14, 15, 25, 41, 61]. Rating-based methods only utilize

user-item interactions for collaborative filtering. KerKT [50] first introduced the kernel induction method with a shadow model for aligning the overlapped users and items. CoNet [17] adopted the cross-connection unit to snitch useful knowledge across domains. DARec [49] applied adversarial training strategy for extracting preference patterns for overlapped users. Supplement-based methods further exploit the auxiliary user/item information, e.g., tags, categories, and reviews etc. TDAR [48] integrated the user-item textual features (e.g., reviews) to transfer useful knowledge. More recently, CFAA [29] further proposed vertical and horizontal attribution alignment between the latent user/item embeddings across domains. However, they cannot better recommendations for cold-start users who do not have rating interactions in the corresponding domain.

**Cold-Start Cross Domain Recommendation.** Cold-Start Cross Domain Recommendation (CSCDR) is set to alleviate the long-standing cold-start problem in recommendation [22, 45]. Existing CSCDR works on this are mainly of two types, i.e., *mapping-based* methods and *meta-based* methods [5, 56]. Mapping-based methods learn to align and transform between the source and target users/items. EMCDR [32] is the most popular method which first adopted the matrix factorization to learn embeddings then utilize a network to bridge users or items from one domain to the other. SSCDR [19] further extends the EMCDR by mapping both users and items with deep metric learning in the semi-supervised manner. Recently, DOML [24] implemented a novel latent orthogonal mapping to extract user preferences over different domains. Meta-based methods integrate the meta learning with the meta network by training on similar tasks. TMCDR [58] took the advantage of the popular Model-Agnostic Meta-Learning (MAML) [12] to optimize a meta network on rating or ranking. PTUPCDR [59] further improved TMCDR with users' characteristic and preference via personalized bridge for modelling. However, the all methods above separated the pre-training part and the mapping part and they cannot be trained end-to-end. Moreover, most of the current approaches cannot better exploit the information among the non-overlapped user-item interactions which also leads to the model degradation.

**Domain Adaptation.** Domain adaptation aims to transfer useful knowledge from the source samples with labels to target samples without labels for enhancing the target performance. Eric Tzeng et al. [42] first implemented maximum mean discrepancy [4] to measure and reduce the domain bias. Baochen Sun et al. [39, 40] further adopted correlation matching via covariance matrix. More recently,

ESAM [8] extended correlation matching with attribution correlation congruence for solving the long-tailed item recommendation problem. Meanwhile, Ganin et al. [13] proposed Domain Adversarial Neural Network (DANN) which utilized a domain discriminator with adversarial training to align the embeddings across domains. Feng Yuan et al. [49] and Wenhui Yu et al. [48] adopted adversarial learning for the cross domain recommendation. Nowadays, more researchers have utilized optimal transport [10, 47], which have the ability of encoding class-structure in distributions for minimizing the global transportation cost. Yitong Meng et al. [33] is the first attempt to apply Wasserstein distance optimal transport for item cold-start recommendation. In this paper, we propose latent distribution alignment for both overlapped and non-overlapped users via unbalance distribution optimal transport to reduce the domain discrepancy.

## 3 MODELING FOR UDMCF

First, we describe notations. We assume there are two domains, i.e., a source domain $\mathcal{S}$ and a target domain $\mathcal{T}$. There are $N_{U_S}$ and $N_{U_T}$ users in source and target domains respectively. There are $N_{V_S}$ and $N_{V_T}$ items in source and target domains respectively. Let $R^{\mathcal{S}} \in \mathbb{R}^{N_{U_S} \times N_{V_S}}$ and $R^{\mathcal{T}} \in \mathbb{R}^{N_{U_T} \times N_{V_T}}$ be the observed source and item rating matrices in $\mathcal{S}$ and $\mathcal{T}$ respectively. To simplify the problem, in this paper, we assume both domains have no other auxiliary information. Meanwhile there are some overlapped users across different domains under the CSCDR settings. We use the overlapped user ratio $\mathcal{K}_u$ to measure how many users are concurrence according to previous researchers [19, 59]. Our purpose for the Dual-CSCDR problem can be included into two main tasks, i.e., (1) **Task1:** recommending source items to target users. (2) **Task2:** recommending target items to source users.

We introduce the overview of our proposed **UDMCF** framework, as is illustrated in Fig. 2. **UDMCF** mainly has two modules, i.e., *rating prediction module* and *distribution alignment module*. The rating prediction module aims to learn user/item distributions with observed user-item interactions via graph neural networks. The distribution alignment module is supposed to reduce the domain discrepancy between the source and target users on both overlapped and non-overlapped. We will introduce these two modules later.

### 3.1 Rating Prediction Module

Firstly, we provide the details of the rating prediction module. For convenience, we use the notations and calculation process in the source domain as an example. For the $i$-th user and the $j$-th item, we define their corresponding one-hot ID vectors as $X_i^{\mathcal{U}_S}$ and $X_j^{\mathcal{V}_S}$, respectively. We adopt a trainable lookup table to exploit the user and item one-hot ID embedding as $\text{LookUp}(X_i^{\mathcal{U}}) = E_i^{\mathcal{U}_S}$ and $\text{LookUp}(X_j^{\mathcal{V}}) = E_j^{\mathcal{V}_S}$. Then we adopt the commonly-used graph neural network to aggregate useful information among the user-item interactions. We first regard the users and items as the nodes in each domain and construct the corresponding graph $A^{\mathcal{S}}$ and $A^{\mathcal{T}}$ based on the rating matrix $R^{\mathcal{S}}$ and $R^{\mathcal{T}}$ as $A^{\mathcal{S}} = \begin{bmatrix} 0 & R^{\mathcal{S}} \\ (R^{\mathcal{S}})^{\top} & 0 \end{bmatrix}, A^{\mathcal{T}} = \begin{bmatrix} 0 & R^{\mathcal{T}} \\ (R^{\mathcal{T}})^{\top} & 0 \end{bmatrix}$. After that we conduct the graph convolution network on both source and target

domains. Graph convolution network can be computed as:

$$\text{GCN}(X, A^{\mathcal{X}} \mid W^{\mathcal{X}}) = (\widetilde{D}^{\mathcal{X}})^{-\frac{1}{2}} \widetilde{A}^{\mathcal{X}} (\widetilde{D}^{\mathcal{X}})^{-\frac{1}{2}} X W^{\mathcal{X}} \quad (1)$$

where $\mathcal{X} = \{\mathcal{S}, \mathcal{T}\}$ denotes the source and target domains. $X$ denotes the input data and $W^{\mathcal{X}}$ denotes the trainable weights. $\widetilde{D}^{\mathcal{X}} = \text{diag}(\widetilde{A}^{\mathcal{X}} 1)$ denotes the degree matrix for the graph $\widetilde{A}^{\mathcal{X}}$ and $\widetilde{A}^{\mathcal{X}} = A^{\mathcal{X}} + I$. Specifically, we adopt $\ell$-th layers of graph convolution network layers to achieve the users/items' mean and covariance of their distribution:

$$[\boldsymbol{\mu}^{U_{\mathcal{X}}}, \boldsymbol{\mu}^{V_{\mathcal{X}}}] = \text{GCN}(\cdots \text{GCN}([E^{U_{\mathcal{X}}}, E^{V_{\mathcal{X}}}], A^{(\mathcal{X})} \mid W_{\mu}^{(\mathcal{X})}) \cdots)$$

$$[\log(\boldsymbol{\sigma}^{U_{\mathcal{X}}})^2, \log(\boldsymbol{\sigma}^{V_{\mathcal{X}}})^2] = \text{GCN}(\cdots \text{GCN}([E^{U_{\mathcal{X}}}, E^{V_{\mathcal{X}}}], A^{(\mathcal{X})} \mid W_{\sigma}^{(\mathcal{X})}) \cdots)$$
$$(2)$$

where $W_{\mu}^{(\mathcal{X})}$ and $W_{\sigma}^{(\mathcal{X})}$ denote two trainable weights for estimating the mean and covariance respectively. $\boldsymbol{\mu}^{U_{\mathcal{X}}}$ and $\boldsymbol{\sigma}^{U_{\mathcal{X}}}$ denote the mean and covariance in the domain $\mathcal{X}$. Since using the single user or item embeddings cannot depict more complicated user-item relationship, we adopt the Gaussian distribution to parameterized user and item distribution. Specifically, the Gaussian distribution can capture the learning more accuracy relationship between the users and items [18, 31, 34, 38]. Therefore, we can obtain the user/item latent distribution $\mathbb{P}(U^{\mathcal{X}})$ and $\mathbb{P}(V^{\mathcal{X}})$ as follows:

$$\mathbb{P}(U^{\mathcal{X}}) = \mathcal{N}(\boldsymbol{\mu}^{U_{\mathcal{X}}}, (\boldsymbol{\sigma}^{U_{\mathcal{X}}})^2), \quad \mathbb{P}(V^{\mathcal{X}}) = \mathcal{N}(\boldsymbol{\mu}^{V_{\mathcal{X}}}, (\boldsymbol{\sigma}^{V_{\mathcal{X}}})^2) \quad (3)$$

After we obtain the user/item latent distribution, we should train the model based on user-item ratings. To better achieve this goal, we propose the distribution-based metric learning loss with self-adaptive margin as given below:

$$L_R = \sum_{(U_i^{\mathcal{X}}, V_j^{\mathcal{X}}) \in O_P^{\mathcal{X}}} \left[ d_W(\mathbb{P}(U_i^{\mathcal{X}}), \mathbb{P}(V_j^{\mathcal{X}})) - m_i^{\mathcal{X}} \right]_+ $$
$$+ \sum_{(U_i^{\mathcal{X}}, V_k^{\mathcal{X}}) \in O_N^{\mathcal{X}}} \left[ m_i^{\mathcal{X}} - d_W(\mathbb{P}(U_i^{\mathcal{X}}), \mathbb{P}(V_k^{\mathcal{X}})) \right]_+, \quad (4)$$

where $O_P^{\mathcal{X}}$ and $O_N^{\mathcal{X}}$ denote the positive and negative user-item pairs respectively. $[\cdot]_+$ denotes the operation as $[x]_+ = \max(0, x)$. $m_i^{\mathcal{X}}$ denotes self-adaptive margin for the $i$-th user in domain $\mathcal{X}$. We adopt a fully-connected network $G_m^{\mathcal{S}}$ and $G_m^{\mathcal{T}}$ to obtain the adaptive margin as $m_i^{\mathcal{X}} = G_m^{\mathcal{X}}(E_i^{U_{\mathcal{X}}})$. The $d_W(\cdot)$ denotes Wasserstein distance among different Gaussian distributions which can be calculated as:

$$d_W(\mathbb{P}(U_i^{\mathcal{X}}), \mathbb{P}(V_j^{\mathcal{X}})) = d_W(\mathcal{N}(\boldsymbol{\mu}_i^{U_{\mathcal{X}}}, (\boldsymbol{\sigma}_i^{U_{\mathcal{X}}})^2), \mathcal{N}(\boldsymbol{\mu}_j^{V_{\mathcal{X}}}, (\boldsymbol{\sigma}_j^{V_{\mathcal{X}}})^2))$$

$$= ||\boldsymbol{\mu}_i^{U_{\mathcal{X}}} - \boldsymbol{\mu}_j^{V_{\mathcal{X}}}||_2^2 + ||\boldsymbol{\sigma}_i^{U_{\mathcal{X}}} - \boldsymbol{\sigma}_j^{V_{\mathcal{X}}}||_2^2$$

After adopting the metric-based rating prediction loss, we can pull the positive user-item pairs while push away the negative user-item pairs. Meanwhile for the different users, we provide adaptive margins to better pursuit the user preferences.

### 3.2 Distribution Alignment Module

In the common assumptions of CDR, two domains always share similar characteristics [7]. Hence, the user distributions across domains are always similar and should be mapped for transferring useful knowledge. In this section of distribution alignment module, we will provide the details of how to reduce the discrepancy between the source and target domains. We denote $\mathbb{P}_S$ and $\mathbb{P}_T$ as the source and target user probability distributions, respectively. In Dual-CDCSR setting, $\mathbb{P}_S \neq \mathbb{P}_T$ because the user distributions

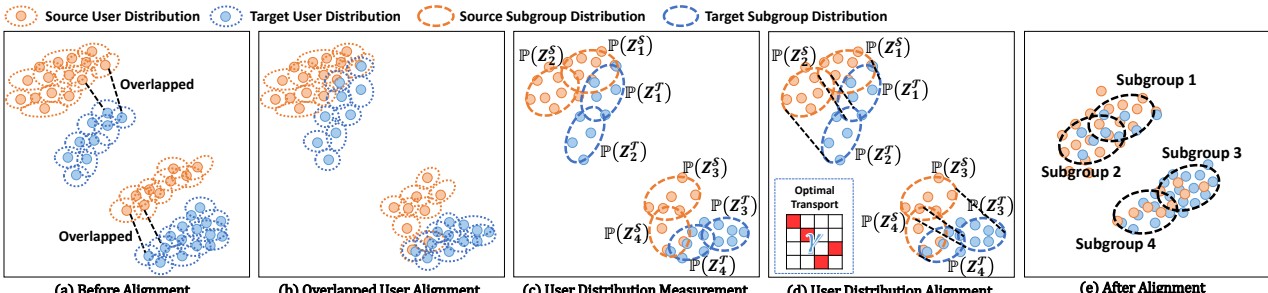

**Figure 3: The basic procedure of distribution alignment module.**

generated from the source domain and the target domain are heterogeneous, which leads to the *domain discrepancy* problem as shown in Fig. 3(a). Specifically, the orange-colored source user distribution and blue-colored target user distribution are separated with the existence of domain bias and discrepancy. Thus, how to reduce the domain discrepancy and transfer useful knowledge across domains has become the key to high-performance Dual-CDCSR. To fulfill this task, we propose a novel distribution alignment module. In the feature space, we consider both overlapped and non-overlapped users' distribution information. Distribution alignment module has two main components, i.e., *Overlapped User Distribution Alignment*, and *General User Subgroup Distribution Alignment* as shown in Fig.2. Overlapped user distribution alignment tends to align the concurrent users across domains to obtain domain-invariant representations. General user distribution alignment tends to map the subgroups user distributions for whole users on source and target domains for knowledge sharing.

*3.2.1 Overlapped User Distribution Alignment.* Intuitively, the overlapped users should share similar latent distributions since they have similar preferences and characteristics. Therefore, we propose the user distribution mapping method for aligning the overlapped users among the $U^S$ and $U^T$. We adopt the Wasserstein distance for the loss function of user embedding mapping as:

$$L_{OU} = \sum_{i=1}^{N} \sum_{k=1}^{N} \delta(U_i^S, U_k^T) \cdot d_W(\mathbb{P}(U_i^S), \mathbb{P}(U_k^T)) \qquad (5)$$

where $\delta(\cdot)$ indicates whether a user occurs simultaneously in both domains under the Dual-CSCDR setting. $\delta(U_i^S, U_k^T) = 1$ means the $i$-th user in the source domain has the same identification with the $k$-th user in the target domain, otherwise $\delta(U_i^S, U_k^T) = 0$. The loss function can further transfer the useful knowledge among the overlapped users to make the representations more expressive as shown in Fig. 3(b). Specifically, the overlapped users in source and target domains can be aligned in the latent space.

*3.2.2 General User Subgroup Distribution Alignment.* Although aligning the overlapped users can enhance the model performance on Dual-CSCDR, it still neglect the abundant knowledge hidden behind the non-overlapped users. The smaller the overlapped ratio $\mathcal{K}_u$ is, the less information could the overlapped users provided. Meanwhile, only aligning the user distributions among the overlapped users cannot directly reduce the domain discrepancy for the whole users as shown in Fig. 3(b). Therefore, it is crucial to figure out the high efficient method to exploit the knowledge for the large majority of non-overlapped users across domains. In order to

better fulfill the task of distribution alignment for the whole users across domains, we innovatively propose latent subgroup distribution alignment method which includes two main steps, i.e., *general user subgroup distribution measurement* for estimating the latent user distribution among the source and target domains, and *general user subgroup distribution adaptation* for reducing the domain discrepancy via optimal transport.

**General User Subgroup Distribution Measurement.** We first assume that both source and target domains have $M$ subgroups $\mathbb{P}(Z^S)$ and $\mathbb{P}(Z^T)$. Meanwhile, users in the corresponding subgroups may have similar tastes or characteristics. Then we should exploit the relationship between these users and their corresponding subgroups. Therefore, we set $\zeta^X \in \mathbb{R}^{N \times M}$ be the similarity matrix between the users and different subgroups. To make things simple, we adopt $X = \{S, T\}$ to represent the source or target domain. We also prefer the similar users should have the similarity value on $\zeta^X$. Inspired by [2, 28, 36], we propose Typical Subgroup Discovering algorithm (TSD) for user distribution measurement with the proximal term as:

$$\min_{\mathbb{P}(Z^X), \zeta^X \in \Delta} \sum_{i=1}^{N} \sum_{j=1}^{M} \left[ \zeta_{ij}^X \mathrm{KL}(\mathbb{P}(U_i^X)||\mathbb{P}(Z_j^X)) + \epsilon_U \cdot \zeta_{ij}^X \log \zeta_{ij}^X \right] \qquad (6)$$

where $\Delta = \{ \zeta_{ij}^X \geq 0, \sum_{j=1}^{M} \zeta_{ij} = 1 \}$ denotes the subjective condition and $\mathbb{P}(Z_j^X) = \mathcal{N}(\mu_j^{Z_X}, (\sigma_j^{Z_X})^2)$ denotes the $j$-th subgroup normal distribution and $\mu_j^{Z_X}, \sigma_j^{Z_X}$ denote the corresponding mean and variance. The second term $\sum_{i=1}^{N} \sum_{j=1}^{M} \zeta_{ij}^X \log \zeta_{ij}^X$ is the entropy regularization for achieving nonnegative and nonlinearly results on $\zeta^X$ [2].

**Optimization.** We neglect the irrelevant constant terms in Eq.(6) and use Lagrangian multiplier to minimize the objective function:

$$\min_{\mathbb{P}(Z^X), \zeta^X} J = \sum_{i=1}^{N} \sum_{j=1}^{M} \left[ \zeta_{ij}^X \sum_{d=1}^{D} \left[ \log \frac{\sigma_{j,d}^{Z_X}}{\sigma_{i,d}^{U_X}} + \frac{(\sigma_{i,d}^{U_X})^2 + (\mu_{i,d}^{U_X} - \mu_{j,d}^{Z_X})^2}{2(\sigma_{j,d}^{Z_X})^2} \right] \right]$$
$$+ \sum_{i=1}^{N} \sum_{j=1}^{M} \epsilon_U \cdot \zeta_{ij}^X \log \zeta_{ij}^X + \sum_{i=1}^{N} \vartheta_i \left( \sum_{j=1}^{M} \zeta_{ij}^X - 1 \right) \qquad (7)$$

where $\vartheta$ is the Lagrangian multiplier for the constraints. We first fix the variable $\mu_j^{Z_X}, \sigma_j^{Z_X}$ and update $\zeta_{ij}^X$. Taking the differentiation of Eq.(7) w.r.t. $\zeta_{ij}^X$ and setting it to 0, we can obtain:

$$\frac{\partial J}{\partial \zeta_{ij}^X} = 0 \Rightarrow \zeta_{ij}^X = \frac{\exp(-\mathrm{KL}(\mathbb{P}(U_i^X)||\mathbb{P}(Z_j^X))/\epsilon_U)}{\sum_{k=1}^{M} \exp(-\mathrm{KL}(\mathbb{P}(U_i^X)||\mathbb{P}(Z_k^X))/\epsilon_U)}, \qquad (8)$$

Then we fix the variable $\sigma_j^{Z_X}$, $\zeta_{ij}^X$ and update $\mu_j^{Z_X}$. Taking the differentiation of Eq. (7) w.r.t. $\mu_j^{Z_X}$ and setting it to 0, we can update $\mu_j^{Z_X}$ via $\mu_{j,d}^{Z_X} = \sum_{i=1}^N \zeta_{ij}^X \mu_{i,d}^{U_X} / \sum_{i=1}^N \zeta_{ij}^X$. Finally we fix the variable $\mu_j^{Z_X}$, $\zeta_{ij}^X$ and update $\sigma_j^{Z_X}$. Taking the differentiation of Eq. (7) w.r.t. $\sigma_j^{Z_X}$ and setting it to 0, we can update $\sigma_j^{Z_X}$ as:

$$\frac{\partial \ell_X}{\partial \sigma_{j,d}^{Z_X}} = 0 \Rightarrow (\sigma_{j,d}^{Z_X})^2 = \frac{\sum_{i=1}^N \zeta_{ij}^X \cdot [(\sigma_{i,d}^{U_X})^2 + (\mu_{i,d}^{U_X} - \mu_{j,d}^{Z_X})^2]}{\sum_{i=1}^N \zeta_{ij}^X}. \quad (9)$$

After several iterations, we can obtain the stable values of $\zeta_{ij}^X$, $\mu_j^{Z_X}$ and $\sigma_j^{Z_X}$. Therefore, the whole user distribution $\mathbb{P}^X(x)$ can be depicted as a combination of these subgroups:

$$\mathbb{P}^X(x) = \sum_{k=1}^M \pi_k^X \mathcal{N}(\mu_k^{Z_X}, \Sigma_k^{Z_X}) = \sum_{k=1}^M \pi_k^X \mathbb{P}(Z_k^X), \pi_k^X = \frac{1}{N}\sum_{i=1}^N \zeta_{ik}^X. \quad (10)$$

Through the calculation above, we can extract the latent user subgroup distribution as shown in Fig. 3(c).

**General User Subgroup Distribution Adaptation.** After obtaining the latent user distribution on the feature space, we tend to reduce the domain discrepancy for transferring useful knowledge among the non-overlapped users. Inspired by the widely used optimal transport techniques based on Kantorovich problem [1], we first proposed unbalanced distribution optimal transport with coupling matrix $\gamma \in \mathbb{R}^{M \times M}$ as:

$$\gamma^* := \arg\min_\gamma \int_{\mathbb{P}^S \times \mathbb{P}^T} C(\mathbb{P}^S, \mathbb{P}^T) \, d\gamma(\mathbb{P}^S, \mathbb{P}^T). \quad (11)$$

The matrix function $C(\mathbb{P}^S, \mathbb{P}^T)$ denotes the cost to move from the source to target user probability distribution. Traditional distribution optimal transport always assume that the components of each subgroup are balance [30, 46]. However, it cannot handle the situation when the proportion of subgroup distributions are unbalance [3]. This situation will always occur, e.g., the majority of users in source domain prefer romantic items while the majority of users in target domain prefer realistic items. At that time, the traditional distribution optimal transport may result in the worse coupling matrix due to the strict constraints across domains [11, 37]. In order to resolve this issue, we propose Unbalance Distribution Optimal Transport (UDOT) by relaxing the original hard constraint through Kulback-Leibler divergence as:

$$\min_\gamma \ell_\gamma = \sum_{i=1}^M \sum_{j=1}^M \gamma_{ij} d_W\left(\mathbb{P}(Z_i^S), \mathbb{P}(Z_j^T)\right) + \epsilon_\gamma \sum_{i=1}^M \sum_{j=1}^M \gamma_{ij}(\log(\gamma_{ij}) - 1)$$
$$+ \tau \text{KL}\left(\gamma \mathbf{1}_M || \pi^S\right) + \tau \text{KL}\left(\gamma^\top \mathbf{1}_M || \pi^T\right), \quad (12)$$

where $d_W(\mathbb{P}(Z_i^S), \mathbb{P}(Z_j^T))$ denotes the Wasserstein distance between the $i$-th and the $j$-th subgroup in the source and target domains respectively. $\epsilon_\gamma$ and $\tau$ denote as the balanced hyper parameters. The $\text{KL}(x||y)$ denotes the KL Divergence between two $d$-dimensional data samples $x \in \mathbb{R}^D$ and $y \in \mathbb{R}^D$ as $\text{KL}(x||y) = \sum_{i=1}^D \left[x_i \log \frac{x_i}{y_i} - x_i + y_i\right]$.

**Optimization.** The UDOT can be solved by taking the differentiation on the coupling matrix $\gamma$ in Eq.(12) to obtain:

$$\frac{\partial \ell_\gamma}{\partial \gamma_{ij}} = d_W(p_i^S, p_j^T) + \epsilon_\gamma \log \gamma_{ij} + \tau \log \frac{\gamma \mathbf{1}_M}{\pi_i^S} + \tau \log \frac{\gamma^\top \mathbf{1}_M}{\pi_j^T} = 0. \quad (13)$$

We apply the new variables $\kappa$ and $\omega$ as:

$$\gamma \mathbf{1}_M = \pi_i^S \exp\left(-\frac{\kappa_i}{\tau}\right), \quad \gamma^\top \mathbf{1}_M = \pi_j^T \exp\left(-\frac{\omega_j}{\tau}\right). \quad (14)$$

Meanwhile the coupling matrix can be depicted as $\gamma_{ij} = \exp((\kappa_i + \omega_j - d_W(\mathbb{P}(Z_i^S), \mathbb{P}(Z_j^T)))/\epsilon_\gamma)$. Taking them back to the Eq.(12), we could obtain the Fenchel-Lagrange conjugate form of the UDOT as:

$$\min_{\kappa, \omega} \ell_u = \tau \sum_{i=1}^M \left[\pi_i^S \exp\left(-\frac{\kappa_i}{\tau}\right)\right] + \tau \sum_{j=1}^M \left[\pi_j^T \exp\left(-\frac{\omega_j}{\tau}\right)\right]$$
$$+ \epsilon_\gamma \sum_{i=1}^M \sum_{j=1}^M \exp\left(\frac{\kappa_i + \omega_j - d_W(\mathbb{P}(Z_i^S), \mathbb{P}(Z_j^T))}{\epsilon_\gamma}\right). \quad (15)$$

The UDOT problem can be effectively solved by alternatively updating $\kappa$ and $\omega$. Therefore, we take the differentiation w.r.t. on $\kappa$, $\omega$ and set it equals to 0 as follows:

$$\begin{cases} \kappa_i = \frac{\tau \cdot \epsilon_\gamma}{\tau + \epsilon_\gamma}\left(\log(\pi_i^S) - \log\left(\sum_{j=1}^M \exp\left(\frac{\omega_j - d_W(\mathbb{P}(Z_i^S), \mathbb{P}(Z_j^T))}{\epsilon_\gamma}\right)\right)\right) \\ \omega_j = \frac{\tau \cdot \epsilon_\gamma}{\tau + \epsilon_\gamma}\left(\log(\pi_j^T) - \log\left(\sum_{i=1}^M \exp\left(\frac{\kappa_i - d_W(\mathbb{P}(Z_i^S), \mathbb{P}(Z_j^T))}{\epsilon_\gamma}\right)\right)\right) \end{cases} \quad (16)$$

After several iterations, we will achieve the optimal solution of $\kappa_i^*$ and $\omega_j^*$. Then we can obtain accurate coupling matrix $\gamma^*$ shown in Fig. 3(d). Finally, the loss of user distribution alignment can be provided as follows:

$$L_{NU} = \sum_{i=1}^M \sum_{j=1}^M \exp\left(\frac{\kappa_i^* + \omega_j^* - d_W(\mathbb{P}(Z_i^S), \mathbb{P}(Z_j^T))}{\epsilon_\gamma}\right) \cdot d_W(\mathbb{P}(Z_i^S), \mathbb{P}(Z_j^T)) \quad (17)$$

Utilizing the user distribution alignment loss, we can align the whole user distributions across domains shown in Fig. 3(e). Note that even if only few users are overlapped, we can reduce the domain bias and discrepancy for knowledge sharing.

## 3.3 Overall Procedure

The total loss of **UDMCF** could be obtained by combining the losses of the rating prediction module and the distribution alignment module. That is, the loss of **UDMCF** is given as $\min L_{\text{UDMCF}} = L_R + \lambda_{OU}L_{OU} + \lambda_{NU}L_{NU}$, where $\lambda_{OU}$ and $\lambda_{NU}$ are hyper-parameters to balance different types of losses. By doing this, users with similar preference will be gathered across domains as shown in Fig.3(e). In testing phase for solving the **Task1**, one can predict the ratings between the source items and target users by taking the inner product $\langle U^T, V^S \rangle$. Similarly for the **Task2**, one can predict the ratings between the source users and target items by taking the inner product $\langle U^S, V^T \rangle$. Furthermore, we provide the the time complexity of our proposed method. Specifically, the typical subgroup discovering algorithm in user subgroup distribution measurement has the time complexity of $O(N \times M)$. Then the time complexity of unbalance distribution optimal transport in user subgroup distribution alignment is $O(M \times M)$. Finally, the total time complexity of proposed **UDMCF** is $O(N \times M + M \times M)$.

## 4 EMPIRICAL STUDY

In this section, we conduct experiments on several real-world datasets to answer the following questions: (1) **RQ1**: How does our approach perform compared with the state-of-the-art recommendation methods? (2) **RQ2**: How do the overlapped user distribution

alignment and general user distribution alignment contribute to performance improvement on different value of $\mathcal{K}_u$? (3) **RQ3**: How does the performance of **UDMCF** vary with different values of the hyper-parameters?

## 4.1 Datasets and Tasks

We conduct extensive experiments on two popularly used real-world datasets, i.e., *Douban* and *Amazon*. The **Douban** dataset [53, 55] has two domains, i.e., Movie ans Book. The **Amazon** dataset [35, 51] has three domains, i.e., Movies and TV (Movie), Books (Book), and CDs and Vinyl (Music). The detailed statistics of these datasets have be shown in Table 1. For both datasets, we binarize the ratings to 0 and 1. Specifically, we take the ratings higher or equal to 4 as positive and others as 0. We provide three main scenarios to evaluate our model, i.e., **Douban Movie** & **Douban Book**, **Amazon Movie** & **Amazon Music**, and **Amazon Book** & **Amazon Music**. It is noticeable that each scenarios includes the two main tasks under Dual-CSCDR settings.

## 4.2 Experiment Settings

We randomly divide the user-item rating data into training, validation, and test sets with a ratio of 8:1:1. Meanwhile, we vary the overlapped user ratio $\mathcal{K}_u$ in $\{5\%, 50\%, 90\%\}$. We adopt the same method to adjust the overlapped user ratios following previous works [19]. In practice, we first choose the overlapped user according to the overlapped user ratio which was given. Then we keep half of the non-overlapped users across domains. The rest of non-overlapped users-item interactions are removed in the training phase and they can be regarded as the cold-start users. These cold-start users-item interactions are evaluated in the testing phase. Different user overlapped ratio represents different situations, e.g., $\mathcal{K}_u = 5\%$ represents only few users are overlapped while $\mathcal{K}_u = 90\%$ means most of users are overlapped following previous researches [19, 50]. We set batch size $N = 256$ for training. The latent dimension of mean/covariance for users and items are set to $D = 128$. In the rating prediction module, we set the number of graph convolution layers as $\ell = 3$. We set the latent user subgroup clusters as $M = 20$ in user distribution measurement. The entropy regularization term is set as $\epsilon_U = 1$ for user distribution measurement. For the unbalanced distribution optimal transport, we set $\epsilon_\gamma = 1$ and $\tau = 3$. For **UDMCF** model, we set the balance hyper-parameters as $\lambda_{OU} = 0.5$ and $\lambda_{NU} = 0.5$ empirically. In practice, we first choose the overlapped user according to the overlapped user ratio which was given. Then we keep half of the non-overlapped users across domains. The rest of non-overlapped users-item interactions are removed in the training phase and they can be regarded as the cold-start users. For all the experiments, we perform five random experiments and report the average results. We choose Adam [20] as optimizer, and adopt Hit Rate@$k$ (HR@$k$) and NDCG@$k$ [44] as the ranking evaluation metrics with $k = 10$.

## 4.3 Baseline

We compare our proposed **UDMCF** with the following state-of-the-art recommendation models. (1) **NeuMF** [16] is the most popular recommendation model which utilizes the neural network for collaborative filtering on the single domain. (2) **EMCDR** [32] is the

**Table 1: Statistics on Douban and Amazon datasets.**

| Datasets | | Users | Items | Ratings | Density |
|---|---|---|---|---|---|
| **Douban** | Movie | 29,476 | 24,091 | 591,258 | 0.08% |
| | Book | | 41,884 | 579,131 | 0.05% |
| **Amazon** | Movie | 15,914 | 17,794 | 416,228 | 0.14% |
| | Music | | 20,058 | 280,398 | 0.09% |
| **Amazon** | Book | 16,267 | 18,467 | 233,251 | 0.08% |
| | Music | | 21,054 | 195,550 | 0.07% |

popular CSCDR model which utilizes neural network to bridge the user embeddings from the source to target domains based on the matrix factorization. (3) **DCDCSR** [54] utilized the sparsity degrees of individual users and items to guide the collaborative filtering and the mapping process across domains. (4) **SSCDR** [19] adopts a semi-supervised approach for metric space mapping and multi-hop neighborhood inference. (5) **LACDR** [43] adopts the dual autoencoder framework to align the overlapped users in the latent embedding space for cold-start recommendation. (6) **TMCDR** [58] proposes a transfer-meta framework with a transfer stage and a meta stage with matrix factorization. (7) **DOML** [24] is the state-of-the-art cross-domain method which adopts dual metrics learning in cross-domain recommendation. (8) **BiTGCF** [27] adopts the graph neural network with feature transfer layer to fuse users representations for solving the CSCDR problem (9) **PTUPCDR** [59] utilizes a meta network fed with users' characteristic embeddings for personalized preferences transfer. (10) **CDRIB** [6] is the state-of-the-art model for CSCDR which adopts the graph-based information bottleneck to derive user/item unbiased representations. Besides, for a fair comparison, all the models use the same types of data and pre-processing methods during experiments.

## 4.4 Recommendation Performance (for RQ1)

**Results and discussion.** The comparison results on Douban and Amazon datasets are shown in Table 2. The superscript of (S) and (T) on the corresponding dataset represent the source and target domains respectively. Meanwhile the superscript of (T1) and (T2) on the corresponding evaluation metric (i.e., HR and NDCG) indicate the Task1 and Task2 respectively. From them, we can find that: (1) Single domain recommendation model (e.g., **NeuMF**) cannot provide satisfactory results on the Dual-CSCDR problem since it cannot reduce the discrepancy between the source and target domains. Therefore, it is essential to map and transfer useful information across domains. (2) Cold-start cross domain recommendation models (e.g., **EMCDR**) provides better results when the overlapped user ratio is relatively high ($\mathcal{K}_u = 90\%$). However, the recommendation performance degraded when the overlapped user ratio is relatively small ($\mathcal{K}_u = 5\%$). At that time, only very few knowledge can be transferred among these overlapped user. (3) Better modelling the latent user distributions and mapping function (e.g., utilizing the orthogonal weights on the transformation in **DOML**) can indeed improve the representation ability on the Dual-CSCDR task. Nevertheless, they cannot fully exploit the structure of the whole latent distributions on the feature space which limits their potentials. (4) Furthermore, all previous CSCDR methods separated the modelling and mapping process which finally leading to the insufficient performance caused by the error of superimposition. (5) **UDMCF** with distribution alignment module can further enhance the performance under the Dual-CSCDR settings. Moreover,

**Table 2: Experimental results on Douban and Amazon datasets.**

| (Douban) Movie$^{(S)}$ & Book$^{(T)}$ | $\mathcal{K}_u = 5\%$ | | | | $\mathcal{K}_u = 50\%$ | | | | $\mathcal{K}_u = 90\%$ | | | |
|---|---|---|---|---|---|---|---|---|---|---|---|---|
| | HR$^{T1}$ | NDCG$^{T1}$ | HR$^{T2}$ | NDCG$^{T2}$ | HR$^{T1}$ | NDCG$^{T1}$ | HR$^{T2}$ | NDCG$^2$ | HR$^{T1}$ | NDCG$^{T1}$ | HR$^{T2}$ | NDCG$^{T2}$ |
| NeuMF | .1612 | .0706 | .1492 | .0581 | .1987 | .1143 | .2097 | .0989 | .2390 | .1441 | .2276 | .1105 |
| EMCDR | .1798 | 0893 | .1654 | .0763 | .2206 | .1268 | .2239 | .1110 | .2477 | .1525 | .2342 | .1193 |
| DCDCDR | .1820 | .0936 | .1686 | .0802 | .2258 | .1304 | .2275 | .1145 | .2519 | .1560 | .2409 | .1221 |
| SSCDR | .1851 | .0974 | .1705 | .0841 | .2315 | .1332 | .2272 | .1130 | .2558 | .1547 | .2462 | .1264 |
| TMCDR | .1873 | .1002 | .1720 | .0875 | .2397 | .1389 | .2316 | .1204 | .2602 | .1596 | .2514 | .1285 |
| LACDR | .1909 | .1022 | .1713 | .0887 | .2405 | .1391 | .2330 | .1176 | .2643 | .1615 | .2527 | .1298 |
| DOML | .1925 | .1045 | .1762 | .0916 | .2384 | .1437 | .2341 | .1192 | .2667 | .1642 | .2553 | .1320 |
| BiTGCF | .1924 | .1052 | .1781 | .0920 | .2419 | .1448 | .2367 | .1215 | .2703 | .1631 | .2578 | .1349 |
| PTUPCDR | .1912 | .1063 | .1789 | .0904 | .2420 | .1431 | .2363 | .1223 | .2736 | .1634 | .2590 | .1356 |
| CDRIB | .1953 | .1097 | .1827 | .0936 | .2445 | .1459 | .2394 | .1250 | .2761 | .1662 | .2628 | .1375 |
| **UDMCF**-Base | .1804 | .0961 | .1713 | .0862 | .2362 | .1407 | .2294 | .1185 | .2524 | .1553 | .2481 | .1262 |
| **UDMCF**-Overlapped | .1946 | .1083 | .1825 | .0930 | .2453 | .1464 | .2382 | .1247 | .2755 | .1678 | .2634 | .1396 |
| **UDMCF**-ESAM | .2177 | .1240 | .2094 | .1091 | .2609 | .1579 | .2506 | .1353 | .2798 | .1721 | .2702 | .1453 |
| **UDMCF**-DOT | .2215 | .1229 | .2123 | .1084 | .2578 | .1552 | .2530 | .1376 | .2823 | .1735 | .2691 | .1478 |
| **UDMCF** | **.2331** | **.1398** | **.2269** | **.1215** | **.2702** | **.1646** | **.2615** | **.1457** | **.2914** | **.1792** | **.2778** | **.1543** |

| (Amazon) Movie$^{(S)}$ & Music$^{(T)}$ | $\mathcal{K}_u = 5\%$ | | | | $\mathcal{K}_u = 50\%$ | | | | $\mathcal{K}_u = 90\%$ | | | |
|---|---|---|---|---|---|---|---|---|---|---|---|---|
| | HR$^{T1}$ | NDCG$^{T1}$ | HR$^{T2}$ | NDCG$^{T2}$ | HR$^{T1}$ | NDCG$^{T1}$ | HR$^{T2}$ | NDCG$^{T2}$ | HR$^{T1}$ | NDCG$^{T1}$ | HR$^{T2}$ | NDCG$^{T2}$ |
| NeuMF | .0643 | .0315 | .0627 | .0282 | .1012 | .0713 | .1194 | .0728 | .1530 | .0929 | .1581 | .0883 |
| EMCDR | .0771 | .0392 | .0801 | .0357 | .1095 | .0829 | .1280 | .0804 | .1569 | .0993 | .1662 | .0978 |
| DCDCDR | .0819 | .0453 | .0834 | .0389 | .1143 | .0860 | .1327 | .0839 | .1613 | .1028 | .1688 | .1026 |
| SSCDR | .0802 | .0486 | .0864 | .0443 | .1170 | .0896 | .1363 | .0885 | .1639 | .1051 | .1726 | .1012 |
| TMCDR | .0846 | .0521 | .0918 | .0480 | .1215 | .0934 | .1396 | .0867 | .1672 | .1089 | .1763 | .1041 |
| LACDR | .0867 | .0518 | .0943 | .0461 | .1235 | .0959 | .1422 | .0896 | .1683 | .1104 | .1775 | .1060 |
| DOML | .0885 | .0540 | .0956 | .0472 | .1264 | .0978 | .1432 | .0923 | .1714 | .1122 | .1795 | .1087 |
| BiTGCF | .0910 | .0563 | .0935 | .0466 | .1281 | .0962 | .1459 | .0920 | .1693 | .1107 | .1792 | .1076 |
| PTUPCDR | .0907 | .0554 | .0942 | .0505 | .1296 | .0971 | .1458 | .0912 | .1709 | .1145 | .1810 | .1094 |
| CDRIB | .0928 | .0575 | .0967 | .0519 | .1330 | .1014 | .1473 | .0936 | .1718 | .1140 | .1832 | .1123 |
| **UDMCF**-Base | .0860 | .0505 | .0878 | .0431 | .1156 | .0944 | .1429 | .0870 | .1575 | .1002 | .1694 | .0990 |
| **UDMCF**-Overlapped | .0934 | .0587 | .0995 | .0524 | .1348 | .1030 | .1486 | .0951 | .1731 | .1158 | .1846 | .1143 |
| **UDMCF**-ESAM | .1192 | .0776 | .1227 | .0713 | .1442 | .1096 | .1624 | .1019 | .1820 | .1211 | .1908 | .1197 |
| **UDMCF**-DOT | .1178 | .0793 | .1250 | .0726 | .1469 | .1085 | .1602 | .1034 | .1816 | .1223 | .1891 | .1185 |
| **UDMCF** | **.1304** | **.0912** | **.1391** | **.0844** | **.1595** | **.1176** | **.1713** | **.1125** | **.1887** | **.1304** | **.1952** | **.1236** |

| (Amazon) Book$^{(S)}$ & Music$^{(T)}$ | $\mathcal{K}_u = 5\%$ | | | | $\mathcal{K}_u = 50\%$ | | | | $\mathcal{K}_u = 90\%$ | | | |
|---|---|---|---|---|---|---|---|---|---|---|---|---|
| | HR$^{T1}$ | NDCG$^{T1}$ | HR$^{T2}$ | NDCG$^{T2}$ | HR$^{T1}$ | NDCG$^{T1}$ | HR$^{T2}$ | NDCG$^{T2}$ | HR$^{T1}$ | NDCG$^{T1}$ | HR$^{T2}$ | NDCG$^{T2}$ |
| NeuMF | .0472 | .0310 | .0595 | .0269 | .1124 | .0543 | .1237 | .0606 | .1395 | .0707 | .1518 | .0781 |
| EMCDR | .0689 | .0428 | .0782 | .0398 | .1206 | .0667 | .1314 | .0685 | .1481 | .0860 | .1621 | .0898 |
| DCDCDR | .0726 | .0451 | .0819 | .0434 | .1227 | .0690 | .1335 | .0730 | .1513 | .0889 | .1634 | .0935 |
| SSCDR | .0748 | .0460 | .0844 | .0482 | .1253 | .0725 | .1367 | .0751 | .1544 | .0920 | .1673 | .0972 |
| TMCDR | .0797 | .0482 | .0873 | .0456 | .1271 | .0754 | .1408 | .0790 | .1576 | .0916 | .1696 | .1009 |
| LACDR | .0818 | .0495 | .0902 | .0484 | .1302 | .0771 | .1429 | .0823 | .1594 | .0935 | .1710 | .1022 |
| DOML | .0830 | .0518 | .0915 | .0523 | .1315 | .0789 | .1424 | .0828 | .1608 | .0952 | .1715 | .1036 |
| BiTGCF | .0843 | .0509 | .0922 | .0505 | .1330 | .0815 | .1451 | .0842 | .1637 | .0978 | .1743 | .1055 |
| PTUPCDR | .0865 | .0506 | .0907 | .0511 | .1318 | .0812 | .1459 | .0824 | .1622 | .0975 | .1749 | .1067 |
| CDRIB | .0893 | .0521 | .0941 | .0546 | .1330 | .0840 | .1478 | .0857 | .1646 | .0989 | .1758 | .1069 |
| **UDMCF**-Base | .0711 | .0467 | .0830 | .0475 | .1242 | .0778 | .1414 | .0776 | .1490 | .0868 | .1632 | .0924 |
| **UDMCF**-Overlapped | .0936 | .0561 | .0973 | .0567 | .1351 | .0867 | .1505 | .0878 | .1667 | .1003 | .1770 | .1078 |
| **UDMCF**-ESAM | .1164 | .0630 | .1165 | .0682 | .1459 | .0914 | .1576 | .0945 | .1703 | .1047 | .1798 | .1122 |
| **UDMCF**-DOT | .1152 | .0649 | .1186 | .0675 | .1487 | .0891 | .1593 | .0934 | .1695 | .1030 | .1816 | .1108 |
| **UDMCF** | **.1278** | **.0734** | **.1330** | **.0796** | **.1604** | **.0993** | **.1681** | **.1018** | **.1756** | **.1112** | **.1863** | **.1174** |

we also observe that even the overlapped user ratio $\mathcal{K}_u$ is much smaller (e.g., $\mathcal{K}_u = 5\%$), our proposed **UDMCF** can also have great prediction improvement. It indicates that our proposed method can be suitable to solve the Dual-CSCDR problem.

## 4.5 Analysis (for RQ2 and RQ3)

**Ablation.** To study how does each module of **UDMCF** contribute on the final performance, we compare **UDMCF** with its several variants, including **UDMCF-Base** and **UDMCF-Overlapped**. **UDMCF-Base** only adopts rating prediction module without the distribution alignment module. **UDMCF-Overlapped** integrates overlapped user distribution alignment with rating prediction module. In order to validate the effectiveness of our proposed latent distribution alignment via unbalance distribution optimal transport, we further compare our model with **UDMCF-ESAM** and **UDMCF-DOT**. **UDMCF-ESAM** implements the correlation alignment method in **ESAM** to replace the unbalance distribution optimal transport in general user distribution alignment. **UDMCF-DOT** replaces unbalance distribution optimal transport as traditional distribution

optimal transport as well. The comparison results are shown in Table 1. From it, we can observe that (1) Although **UDMCF-Base** can exceed the single domain recommendation model **NeuMF**, it still cannot provide better results for cross domain recommendation. (2) **UDMCF-Overlapped** boost the recommendation performance especially when the overlapped user ratio is relatively high (e.g.,$\mathcal{K}_u = 90\%$). However, the results decrease when there are only few overlapped users due to the limitation of knowledge sharing. (3) **UDMCF-ESAM** and **UDMCF-DOT** both increase the accuracy which illustrates the efficacy of aligning user distribution. While **ESAM** only coarsely aligns the marginal user probability distribution and **DOT** will suffer from the mismatch problem causing by the different mixture proportions [3]. Overall, the above ablation study demonstrates that our proposed model is effective in solving the Dual-CSCDR problem.

**Visualization.** To show the feature space transferability, we visualize the t-SNE embeddings [21] of the source user embeddings ($U^S$) and the target user embeddings ($U^T$). The results on **Amazon Movie**$^{(S)}$ & **Amazon Music**$^{(T)}$ (first column) and **Amazon**

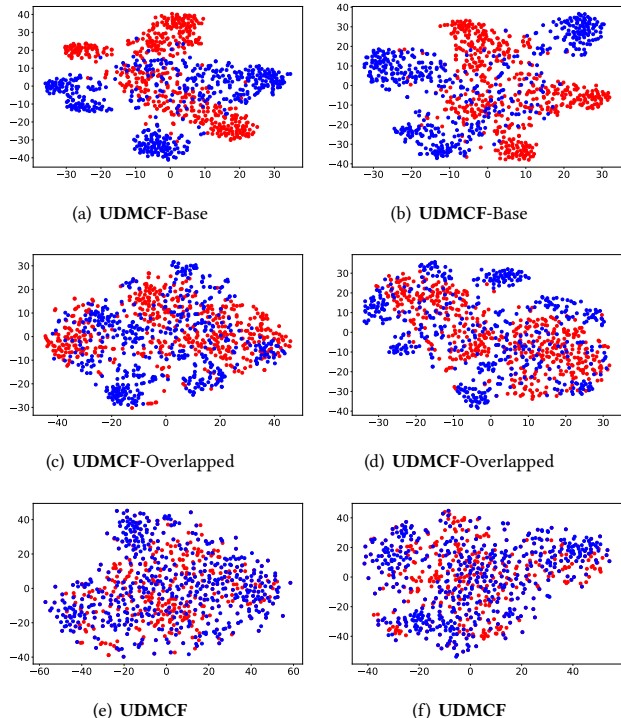

(a) **UDMCF**-Base

(b) **UDMCF**-Base

(c) **UDMCF**-Overlapped

(d) **UDMCF**-Overlapped

(e) **UDMCF**

(f) **UDMCF**

**Figure 4: The t-SNE visualization of user latent embeddings on Amazon Movie$^{(S)}$ & Amazon Music$^{(T)}$ (first column) and Amazon Book$^{(S)}$ & Amazon Music$^{(T)}$ (second column) when the user overlapped ratio is $\mathcal{K}_u = 50\%$. The user latent embeddings in the source domain are shown with red dots and that in the target domain are shown with blue dots.**

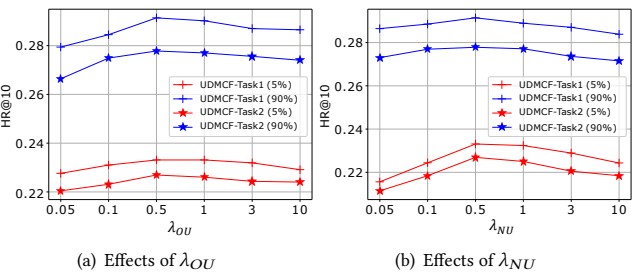

(a) Effects of $\lambda_{OU}$

(b) Effects of $\lambda_{NU}$

**Figure 5: The hyper-parameters of $\lambda_{OU}$ and $\lambda_{NU}$ on Douban Movie$^{(S)}$ & Douban Book$^{(T)}$ with user overlapped ratio $\mathcal{K}_u = 5\%, 90\%$ on Task1 and Task2.**

**Book$^{(S)}$ & Amazon Music$^{(T)}$** (second column) with the user overlapped ratio is $\mathcal{K}_u = 50\%$ are shown in Fig. 4(a)-(f). From it, we can conclude that (1) The domain discrepancy between the source and target users is commonly exist as shown in Fig. 4(a)-(b). Therefore, it is difficult to share and transfer useful knowledge across domains and leading to the poor performance on the Dual-CSCDR problem. (2) Applying **UDMCF**-Overlapped can map and align the overlapped users across domais as shown in Fig. 4(c)-(d). However, the domain bias still exists among the rest non-overlapped users which hurdle the recommendation. (3) Utilizing both overlapped

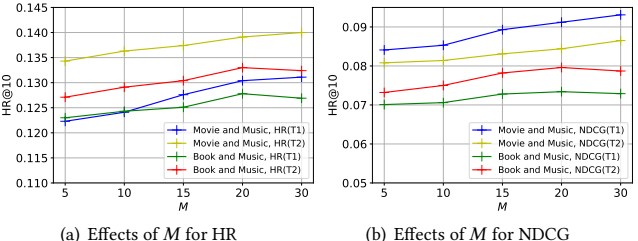

(a) Effects of $M$ for HR

(b) Effects of $M$ for NDCG

**Figure 6: The hyper-parameters of tuning $M$.**

user distribution alignment and general user distribution alignment in **UDMCF** can better align both overlapped and non-overlapped users as shown in Fig. 4(e)-(f). Overall, the above ablation study demonstrates that our proposed distribution alignment module is effective in solving the Dual-CSCDR problem.

**Effect of hyper-parameters.** We finally study the effects of hyper-parameters on model performance with NDCG@10. We vary $\lambda_{OU}$ and $\lambda_{NU}$ in $\{0.05, 0.1, 0.5, 1, 3, 10\}$ on **Douban Movie** & **Douban Book** with user overlapped ratio $\mathcal{K}_u = 5\%$, $\mathcal{K}_u = 90\%$ and report the results in Fig.5(a)-(b). It is straightforward to choose the proper hyper-parameters $\lambda_{OU}$ and $\lambda_{NU}$ to balance the rating prediction and distribution alignment based on the bell-shaped curve in Fig.5. When the $\lambda_{OU}$ and $\lambda_{NU}$ are two small, the distribution alignment loss cannot be adequately trained. However, much larger $\lambda_{OU}$ and $\lambda_{NU}$ may also hinder the training of rating prediction. As a result, we set $\lambda_{OU}$ and $\lambda_{NU}$ equals to 0.5 empirically. Finally, we further tune the number of user subgroups $M$. We vary the $M = \{5, 10, 15, 20, 30\}$ in **Amazon Movie & Music** and **Amazon Book & Music** with $\mathcal{K}_u = 5\%$. Then we report the results of HR and NDCG in Fig.6. The results indicate that fewer subgroups (e.g., $M = 5$ or $M = 10$) may cannot better depict the user general distributions in the latent space. While when number of subgroups are much larger (e.g., $M = 30$), it will cost longer training time and brings about the overfitting in some cases. Therefore, we set $M = 20$ empirically.

## 5 CONCLUSION

In this paper, we propose User Distribution Mapping with Collaborative Filtering (**UDMCF**) for Dual Cold-Start Cross Domain Recommendation, which includes the *rating prediction module* and the *distribution alignment module*. Rating prediction module integrates one-hot ID vector and multi-hot rating interactions for modeling user/item distributions. In distribution alignment module, we innovatively propose overlapped user embedding alignment and general user subgroup distribution alignment to map and transfer knowledge. Specifically, we first propose latent subgroup distribution alignment to measure and align global user distributions across domains. We first propose unbalance distribution optimal transport with typical subgroup discovering to align both overlapped and non-overlapped users. It is noticeable that our proposed **UDMCF** can be trained end-to-end to avoid the error of superimposition. We also conduct extensive experiments to demonstrate the superior performance of our proposed **UDMCF** on several datasets and tasks. In the future, we plan to extend **UDMCF** to more recommendation tasks (e.g., Item cold-start cross-domain recommendation) and conduct more comprehensive experiments on new datasets.

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
