# OpenReview forum: "User Distribution Mapping Modelling with Collaborative Filtering for Cross Domain Recommendation"
_ACM.org/TheWebConf/2024/Conference — TheWebConf24_

### Official Review · Reviewer_Scxp · 2023-11-20

**Novelty:** 5
**Technical Quality:** 5

**Review:**

The author addresses a significant issue in the CDR task, which is how to better utilize non-overlapped data to improve recommendation effectiveness. The author proposes a method using distribution alignment, which is an intriguing approach.

**Questions:**

（1）Methods like graph neural networks can also learn the relationships between non-overlapped users from two different domains through message passing. The author should clearly articulate the advantages of this approach compared to other methods and provide experimental evidence to prove how the proposed method works.
（2）The paper introduces a task called Dual-CSCDR, where in the introduction it is mentioned that it involves recommending source-domain items to source-domain users and target-domain items to target-domain users. This differs from the typical CDR task, but the paper does not provide information about the practical application scenarios of this task. Therefore, readers cannot understand the significance of this task. In the experiments, the author mainly uses baseline models designed for solving CDR tasks, but there is no explanation of how these models can be applied to address the Dual-CSCSR task.

**Reviewer Confidence:**

3: The reviewer is confident but not certain that the evaluation is correct

**Scope:**

4: The work is relevant to the Web and to the track, and is of broad interest to the community

---

### Official Review · Reviewer_nnHC · 2023-11-23

**Novelty:** 6
**Technical Quality:** 6

**Review:**

In this paper, the authors propose UDMCF to integrate the rating prediction module and the feature space alignment module, and verify the effectiveness of the proposed method through a large number of experiments. The framework solves the problem of error superposition in traditional methods by integrating modeling collaborative filtering and feature mapping processes, and makes full use of cross-domain relationships among a large number of non-overlapping users. The framework is novel and the contribution is significant.

Strengths:
•	Dual Cold-Start Cross Domain Recommendation (Dual-CSCDR) problem is relevant to the web conference. Meanwhile it is also an important research problem.
•	On multiple datasets, UDMCF significantly outperforms other state-of-the-art Dual-CSCDR models, demonstrating its excellent performance in this setup.
•	The presentation is good and clear. The model soundness is good as well.

Weakness:
•	Some of the descriptions on the involved parameters may be missing.
•	Some model details should be added.
•	[MINOR] I encourage the authors to provide a pseudo algorithm on the proposed UDOT.

**Questions:**

•	Some of the descriptions on the involved parameters may be missing, for example, how to calculate the KL-divergence term in Eq.(6)?
•	Why should it involve the entropy regularization term in calculating the UDOT in Eq.(12)?
•	[MINOR] I encourage the authors to provide a pseudo algorithm on the proposed UDOT.

**Reviewer Confidence:**

4: The reviewer is certain that the evaluation is correct and very familiar with the relevant literature

**Scope:**

4: The work is relevant to the Web and to the track, and is of broad interest to the community

---

### Official Review · Reviewer_27tx · 2023-11-23

**Novelty:** 6
**Technical Quality:** 6

**Review:**

A dual-cold start cross-domain recommendation method based on user-embedded distributed mapping and Collaborative Filtering (UDMCF) is proposed, which includes two main tasks, namely, recommending source/target items to target/source users. UDMCF consists of two modules: rating prediction module and feature space alignment module. UDMCF aligns the embedding distribution of users across domains by the proposed latent distribution alignment. Extensive experiments on four datasets demonstrate the effectiveness of the proposed method.

Strengths:
 The methodology is technically sound. The proposed typical subgroup discovering algorithm can be rather useful in clustering users with similar preferences. Meanwhile, adopting unbalanced optimal transport for domain adaptation in CDR is novel to me for providing a more accurate matching solution.
 The authors provide the model details including training procedures and the total time complexity information.
 The figures are clear to read. The experimental results show the proposed UDMCF reaches the SOTA performance.

Weaknesses:
 Some experimental details should be more precise.
 Some motivations needed to be explained.

**Questions:**

1. How to adjust the overlapped user ratios during the experiments?
2. Why does the author consider the distribution of users and items in the modeling process? What are the benefits compared to the previous method of using only embedding for modeling?
3. Could a different number of clusters $M$ affect the model performance?

**Reviewer Confidence:**

4: The reviewer is certain that the evaluation is correct and very familiar with the relevant literature

**Scope:**

4: The work is relevant to the Web and to the track, and is of broad interest to the community

---

### Official Review · Reviewer_cxmZ · 2023-11-25

**Novelty:** 5
**Technical Quality:** 5

**Review:**

This paper proposed User Distribution Mapping with Collaborative Filtering (UDMCF) for Dual Cold-Start Cross Domain Recommendation problem. The framework first learns user/item distributions with observed user-item interactions via graph neural networks. Then, it aligns the user representations for both overlapped and non-overlapped users. Extensive experiments demonstrate the effectiveness of the UDMCF.

1.	Quality, clarity, originality, and significant of the work:
The technical aspects of this article are generally sound; the overall expression is clear, but there are a few unclear points; a new problem is proposed, and a new solution is provided; this problem is interesting and has potential.

2.	Strengths:
a)	The problem proposed in this work is interesting and well-motivated.
b)	This the first work aims to align both the overlapped and non-overlapped users embedding distribution across domains.
c)	Proper figures are given to make the paper easy to follow.

3.	Weaknesses:
a)	The detail architecture of the proposed neural network is not given, which seriously reduces the reproducibility.
b)	In section 3.2.2, the authors aim to find a highly efficient method to align the whole users. However, there is no analysis of General User Subgroup Distribution Alignment efficiency.
c)	No code has been released.
d)	There are some minor presentation problems. For example, “Specifically, the Gaussian distribution can capture the learning more accuracy relationship between the users and items [18, 31, 34, 38]” in section 3.1. And notation KL(||) in the equation 6 should be explained at the first occurrence. Moreover, there is a unclarity in section 5, “Specifically, we first propose…” is followed by “We first propose…”.

**Questions:**

1.	The detail architecture of the proposed neural network could be shown in a figure.
2.	There are many components presented in the paper, model complexity is required.
3.	A notation table could help readers to follow the paper.
4.	The “Dual Cold-Start” should be emphasized in the title to align with the content of the article.

**Reviewer Confidence:**

4: The reviewer is certain that the evaluation is correct and very familiar with the relevant literature

**Scope:**

4: The work is relevant to the Web and to the track, and is of broad interest to the community

---

### Official Review · Reviewer_gTdj · 2023-11-26

**Novelty:** 4
**Technical Quality:** 4

**Review:**

Pros:
1.	The paper mainly focuses on the two key points of "cold start" and "cross domain" and proposes the UDMCF model, aiming to solve the Dual CSCDR problem. The main novelty lies in: 1) proposing an end-to-end recommendation framework that integrates the collaborative filtering and distribution mapping process, and 2) fully utilizing the interaction information between overlapping and non-overlapping users and items, achieving distribution alignment to improve recommendation performance. The content of the paper is substantial and of good quality.
2.	The paper is clearly stated and well organized, and the experimental design is relatively reasonable and has good clarity.

Cons：
1.	The methodological novelty of this paper is limited. The rating prediction module of the UDMCF model uses the commonly-used graph neural network, and the methods involved in the distribution alignment module are also mostly traditional classic methods. Moreover, the paper does not state too much about the reasons for the choice of method.
2.	The model framework diagram in the paper is overly simplistic, failing to adequately depict the internal structures of model modules and lacking the model's output.
3.	The model of the paper is only proposed for the Dual-CSCDR problem and does not discuss its potential in other related recommendation problems. The scope of the paper's influence is limited.

**Questions:**

1.	The basis of the paper's design model is mainly around solving cross-domain and cold start problems, but the problem solved in the paper is Dual-CSCDR. Does the model have too little design to solve dual recommendation? Can some methods for dual recommendation be considered?
2.	The paper mainly aligns the distribution from the user's perspective. For example, the General User Subgroup Distribution Adaptation is designed to align the distribution of general users. Is it possible to consider adding a structure such as the General Item Subgroup Distribution Adaptation to achieve the distribution alignment of items?

**Reviewer Confidence:**

3: The reviewer is confident but not certain that the evaluation is correct

**Scope:**

3: The work is somewhat relevant to the Web and to the track, and is of narrow interest to a sub-community

---

### Decision · Program_Chairs · 2024-01-22

**Decision:**

Accept

**Comment:**

A dual-cold start cross-domain recommendation method based on user-embedded distributed mapping and Collaborative Filtering (UDMCF) is proposed, which includes two main tasks, namely, recommending source/target items to target/source users. UDMCF consists of two modules: rating prediction module and feature space alignment module. UDMCF aligns the embedding distribution of users across domains by the proposed latent distribution alignment. Extensive experiments on four datasets demonstrate the effectiveness of the proposed method.

 The authors have sufficiently addressed the questions posed by the reviewers.